# Concomitant Intra-Articular Glenohumeral Lesions in Fractures of the Scapula Body

**DOI:** 10.3390/jcm9040943

**Published:** 2020-03-30

**Authors:** Michael Zyskowski, Sebastian Pesch, Frederik Greve, Markus Wurm, Francesca von Matthey, Daniela Pfeiffer, Sophie Felix, Arne Buchholz, Chlodwig Kirchhoff

**Affiliations:** 1Klinikum rechts der Isar, Technische Universität München, Klinik und Poliklinik für Unfallchirurgie, Ismaninger Str. 22, 81675 München, Germany; Michael.Zyskowski@mri.tum.de (M.Z.); Sebastian.Pesch@mri.tum.de (S.P.); Frederik.Greve@mri.tum.de (F.G.); Wurm@tum.de (M.W.); Francesca.von.matthey@tum.de (F.v.M.); Sophie.Felix@mri.tum.de (S.F.); Arne.Buchholz@mri.tum.de (A.B.); 2Klinikum rechts der Isar, Technische Universität München, Klinik und Poliklinik für Strahlentherapie und Radiologie, Ismaninger Str. 22, 81675 München, Germany; Daniela.Pfeiffer@tum.de

**Keywords:** scapula body fracture, concomitant glenohumeral injuries, level II, prospective cohort study, magnetic resonance imaging, polytrauma, shoulder, rotator cuff

## Abstract

Background: Scapula body fractures are rare injuries with an incidence of 1% of all fractures accounting for 3% to 5% of all upper extremity fractures. Fractures of the scapula commonly result from high-energetic trauma and fall from great height. While several studies focused on concomitant injuries of chest and head as well as the cervical spine, up to now in the common literature, no study exists analyzing the prevalence of concomitant intra-articular glenohumeral injury following extra-articular scapular fracture. Objectives: The aim of this study was to analyze the prevalence of concomitant intra-articular glenohumeral injuries in acute fractures of the scapula by performing magnetic resonance imaging (MRI) of the shoulder joint. Study Design and Methods: This prospective cohort study was performed at our academic Level I trauma center from November 2014 to October 2016. According to our clinical algorithm, all patients suffering from an acute scapula body fracture primarily underwent computed tomography (CT) for assigning the fracture according to the Orthopedic Trauma Association (OTA)-classification and therapy planning. In addition, 3 T MRI-scans of all patients were performed within seven days after trauma. Results: Twenty-one (16 male/5 female, mean age 53 years (25–83 y) patients with scapula body fractures (OTA 14.A3.2 80.1%, OTA 14.A3.1 4.8%, OTA14.B3.1 4.8%, OTA14.C3 9.5%) were enrolled. MRI revealed 11 acute intra-articular injuries in 8 of 21 patients (38%). In all 21 patients, hematoma of the rotator cuff and periarticular muscles was present. Three patients (14.3%) presented a partial bursa sided tear of the supraspinatus tendon, whereas in 5 (23.8%), a partial articular sided supraspinatus tendon tear and in 2 (9.5%) patients, a subtotal tear was observed. One patient (4.8%) showed a complete transmural supraspinatus tendon tear. Conclusions: Traumatic concomitant glenohumeral injuries in scapula body fractures seem to be more frequent than generally expected. Subsequent surgical treatment of these formerly missed but therapy-relevant injuries may increase functional outcome and reduce the postoperative complication rate following scapula body fractures.

## 1. Introduction

The scapula is the conjunction between the arm and the thorax. Since the glenoid articulates with the humerus at the glenohumeral joint, with the clavicle at the acromioclavicular joint, and with the thorax at the so-called scapulothoracic joint, the entire scapula is essential for upper extremity movement. Thus, the range of motion of the shoulder joint strongly depends on the movement of all three joints. Six basic movements of the scapula (elevation, depression, upward rotation, downward rotation, protraction, retraction) are coordinated by 18 different muscle groups, so that a fracture of the scapula can negatively affect the range of motion. 

Fractures (Fx) of the scapula body are of a rare incidence considering all fractures of the upper extremity. In the current literature, small case series showed that 3% to 5% of all Fx of the shoulder girdle affect the scapula, equal to 0.4% to 1% of all fractures [1,2,3,4,5]. High-energy trauma such as high velocity accidents, direct blunt trauma, and fall from great height are the major causes. In 80% to 95%, scapula Fx are associated with additional injuries to the chest such as pneumothorax, pulmonary contusion, but also with pelvic ring fractures, traumatic brain injury (TBI), and splenic or liver lacerations [6,7]. On average, patients with Fx of the scapula suffer from four further injuries [6]. These associated, potentially life-threatening injuries may result in an underestimation of the shoulder injury in thus inappropriate therapy.

Over the last years, several authors have discussed the advantages of surgical treatment of scapula body Fx. The use of new plating systems, minimal invasive methods, and modified approaches resulted in significant better outcome in comparison to conservative treatment [8,9,10,11,12,13,14,15]. 

Besides the fact that scapular Fx are a predictor for a great number of concomitant life-threatening injuries, isolated scapula body Fx have great potential to cause significant pain and to decrease the normal function of the shoulder girdle. Previous research has shown that concomitant intra-articular injuries of the glenohumeral joint in acromioclavicular joint injuries and in lateral clavicula fractures present an incidence of up to 46%, thus should be treated operatively to avoid significant loss of function in the glenohumeral joint [16,17]. 

To the best of our knowledge, there is no study in the literature analyzing the prevalence of concomitant intra-articular glenohumeral injuries in scapula body Fx. Therefore, the purpose of this prospective study was to analyze the prevalence of concomitant intra-articular glenohumeral injuries in patients suffering from acute scapula body Fx by performing magnetic resonance imaging (MRI) of the shoulder joint. 

## 2. Material and Methods

The study protocol was approved by the local ethics committee (Ethics Committee of the Medical Faculty, Technical University of Munich; study number 86/19S) Written informed consent was obtained from each patient. 

Patients suffering from acute scapula body Fx who presented to our university emergency department were included in this prospective cohort study. Medical history was scanned for any known pathologies of the shoulder joint. Patients with a history of glenohumeral pathologies, such as preexisting acute rotator cuff tear, glenohumeral instability, acromioclavicular joint instability, calcifying tendonitis, or biceps tendon pathology, were excluded from the study. 

### 2.1. CT and X-Ray Protocol

Patients suffering from multiple trauma underwent focused assessment with computed tomography in trauma (FACTT) of the whole body [18]. 

Patients suffering from isolated shoulder injury and Fx of the scapula in conventional X-rays underwent additional CT scan of the shoulder, head, and cervical spine.

TBI, abdominal lesions, thoracic injuries, injuries of the pelvic ring, vascular system, and further major and minor injuries of the musculoskeletal system were examined according to advanced trauma life support (ATLS) in two steps. The initial clinical and radiological examination was performed at the emergency department or the trauma shock unit followed by a secondary survey after primary stabilization. A third survey was performed the day after the patients had been admitted to a stationary ward or intensive care unit.

Subsequently, all scapula Fx were classified based on CT according to the Orthopedic Trauma Association (OTA) classification [19,20]. In fracture types OTA 14.A3.2, OTA 14.A3.1, as well as OTA14.C3 and OTA 14.B3.1 with a medial/lateral displacement >20 mm, tilt >45°, displacement >15 mm with tilting >30°, and a glenopolar angle >20° were considered as indications for surgical treatment [21,22].

### 2.2. MRI Protocol 

In the first 14 days post trauma, all patients underwent MRI of the affected shoulder. MRIs were acquired on a 3-Tesla scanner (Ingenia; Philips Healthcare, Best, the Netherlands) using a dedicated 8-channel receive-only shoulder coil. The patients were positioned in a supine position with the shoulder in a neutral position. The imaging protocol comprised fat-saturated proton-density weighted (PDw) sequences in the oblique coronal (parallel to the long axis of the supraspinatus tendon), oblique sagittal (parallel to the glenohumeral joint), and axial plane (echo time, 50 ms; repetition time, 2500 ms; slice thickness, 3 mm; gap, 3 mm; matrix, 500 × 420 mm; field of view, 180–200 mm; flip angle, 90°; pixel bandwidth, 185 Hz/pixel; echo train length, 11), as well as a coronal oblique T1-weighted images (echo time, 15 ms; repetition time, 398 ms; slice thickness, 3 mm; gap, 3 mm; matrix, 520 × 360 mm; field of view, 180 mm; flip angle, 90°; pixel bandwidth, 294 Hz/pixel; echo train length, 7). MRIs were read by an expert musculoskeletal radiologist (DP) and one expert shoulder surgeon (CK) for concomitant intra-articular lesions of the shoulder joint. 

## 3. Results

Between November 2014 and October 2016, 21 patients (16 men/5 women) suffering from scapula body Fx were enrolled. The average age was 53 ± 27 years with an age range between 25 and 83 years. The ratio between left and right scapula body Fx was 8:13. 

Regarding the etiology of scapular Fx, four patients sustained occupational injuries, whereas the remaining 17 Fx were caused by home and leisure accidents. In 12 patients (57.1%), blunt trauma to the chest and spine due to falls from a height were the main causes for scapula body Fx. Three patients (14.3%) had been falling down the stairs. Nine patients (42.8%) suffered from traffic, majorly motorcycle accidents (4 = 19%). Two scapula Fx were found in bikers (9.5%), one in a quadbike driver (4.8%), one in a car driver (4.8%), as well as one in a pedestrian (4.8%) who was hit by a car and pinched with his back between the car and a light pole (see Figure 1). Three (14.3%) patients were admitted to our intensive care unit (ICU) for observation for at least 24 h after trauma. 

### 3.1. Injury Pattern

Initial whole-body trauma CT was only performed in 6 patients. However, in patients who did not undergo initial whole-body CT, a CT scan of the brain and cervical spine was performed. If clinically necessary, additional CT scans of the region of interest were supplemented.

Regarding further injuries to the head, neck, chest, and abdomen, the clinical examination along with the whole-body trauma CT revealed four Fx of the clavicle (19.04%) ipsilateral to the scapula Fx (two type A and two type B- fracture) being treated by open reduction and internal fixation. Eight patients (38.1%) suffered from traumatic brain injury (TBI) with 5 patients suffering from a mild TBI (23.32%) as defined by the American Congress of Rehabilitation Medicine (ACRM, 1993) [23]. Three patients presented with moderate TBI (14.28%) with one case of intracranial hemorrhage, one subarachnoid hemorrhage (4.76%), one subdural hematoma (4.76%), and one cerebral contusion (4.76%). In 6 patients, Fx of the skull and facial bones were identified (28.6%). Two zygomatic arch- (9.52%), one os sphenoidal (4.76%), one orbital blow-out (4.76%), one orbital roof Fx (4.76%), and one (4.76%) Fx of the anterior skull base were detected.

Nine cases of thoracic injuries were found (42.8%). Multiple rip Fx were present in 8 patients (38.1%), all ipsilaterally to the scapula body Fx with a range between 2 and 7 ribs being affected. In addition, 3 patients showed an ipsilateral pneumothorax (14.28%) and 4 presented with ipsilateral pulmonary contusions (19%). One sternum fracture along with mediastinal emphysema and hematoma (4.8%) was identified.

The spine was additionally injured in 5 cases (23.8%), once affecting the cervical spine (4.8%) and, three times the thoracic spine (14.3%). In two of the thoracic spine injuries (9.5%), Th 11 was affected classified as AO A3.3.1 fractures. One patient (4.8%) showed a severe thoracic spine injury involving Th3-Th6 (AO B1.1). All three thoracic spine injuries underwent surgical treatment in terms of dorsal stabilization. One patient presented a fracture of the lumbar spine (4.76%) treated conservatively.

Overall, one Hoffa Type medial femur condyle Fx (4.8%) was detected and treated by Open Reduction Internal Fixation (ORIF) using cannulated 7.0 mm screws. One Fx of the anterior process of the calcaneus (A82.A1.1, 4.8%) was detected and treated conservatively in an immobilizing orthosis with partial weightbearing using crutches. Only one patient had a singular scapula body Fx without any further concomitant injuries (4.8%) (for injury pattern, see Figure 2).

### 3.2. Fracture Pattern

According to the OTA classification, there were 17 14.A3.2 Fx (80.1%), one 14.A3.1 (4.8%) Fx, one 14 B3.1 (4.8%) Fx, as well as two 14 C3 (9.5%) Fx.

According to the indication of surgery, there were six patients (28.6%) with severe displacement of the scapula body consecutively undergoing an open reduction and internal fixation (ORIF) of the scapula body. In all patients, the Arthrex^®^ clavicle fracture plate system was used (see Figure 3).

### 3.3. Concomitant Glenohumeral Lesions

The mean interval between trauma and MR imaging was 6.9 days (0–14 days). In 8 of 21 patients (38%), we found 11 concomitant intra-articular glenohumeral lesions. In three patients (14.3%), two lesions occurred at the same time. In total, three partial bursa sided tears of the supraspinatus tendon (14.3%) and five (23.8%) partial articular sided supraspinatus tendon tears were found. Two patients (9.5%) had a partial tear of the subscapularis tendon Fox and Romeo I (9.5%), whereas only one (4.8%) complete transmural supraspinatus tendon tear was detected (see Figure 4). In none of the detected concomitant intra-articular glenohumeral injuries, a fatty degeneration or/and atrophy of the rotator cuff musculature was found. Moreover, indirect signs of acute rotator cuff injury were found in terms of the 1st bone bruise of the lesser and respectively the greater tuberosity and 2nd joint effusion in terms of acute hematoma; so that the injuries were considered as an acute traumatic tear (see Figure 5). 

In all 21 (100%) patients, MRI showed hematoma of the rotator cuff and periarticular muscles. The MR images allowed for a good to excellent allocation of the hematoma, so that 19 hematomas were detected in the subscapularis muscle (90.5%), followed by 14 hematomas in the infraspinatus muscle (66.6%), 8 patients with hematoma in the teres minor (38.1%), and 5 hematomas in the teres major muscle (23.8%) respectively. One patient had severe intramuscular bleeding of the deltoid muscle (4.8%). In addition, four patients (19%) presented with a fluid filled subacromial bursa. Arthritis of the acromioclavicular joint was detected in 5 cases (23.8%), whereas acute edema of the AC-joint was recognized in three (14.3%) patients. One patient presented with an os acromiale (4.8%), as well as one intramuscular ganglion was found in the infraspinatus muscle. Furthermore, in one patient, signs of reconstruction of the supraspinatus tendon along with previous tenodesis of the long head of the biceps tendon (4.8%) were present, whereas one patient had undergone reconstruction of the supraspinatus tendon only (4.8%). Only a weak correlation was found regarding patient’s age and the presence of a concomitant glenohumeral lesion (Pearson´s correlation coefficient; r = 0.34, see Table 1).

## 4. Discussion

Fractures of the scapular body are rather rare injuries considering all upper extremity Fx, very commonly occurring in multiple trauma patients. Since the scapula is the binding part between the thorax and the upper extremity, concomitant injuries of the shoulder joint are rational. In our study, we were able to present for the first time an assessment of concomitant glenohumeral injuries in scapula body Fx performing MRI. All 21 enrolled patients presented hematoma of the rotator cuff and periarticular muscles, 8 patients showed an either partial bursa or articular sided supraspinatus tendon tear as well as one complete transmural supraspinatus tendon tear. Overall MRI revealed two cases with indication for surgery additional to the scapula Fx repair. 

The common literature reports that scapula Fx occur in 3–5% of all Fx of the shoulder girdle [1]. According to the OTA classification, 17 out of the enrolled 21 patients had an OTA 14.A3.2 fracture (80.1%), 1 (4.8%) suffered from an OTA 14.A3.1 type Fx, one OTA 14.B3.1 Fx was detected (4.8%) and two FX type OTA14.C3 (9.5%). The average age of the presented patient cohort was 53 years (25–83 years), whereas in studies of the current literature on concomitant lesions of the shoulder [3,6,7], the average age is 35 years. However, it should be stated that already a rising trend of fragility Fx in older patients has been observed [24,25] so that the mean age of patients with scapula body Fx will also rise in the next years, which states our patients’ age reasonably even though not in context with scientific literature. 

MRI showed concomitant intra-articular glenohumeral injuries in 8 patients (38%), whereas in two cases (9.5%), the indication for additional arthroscopic treatment of partial thickness tear of tendons of the rotator cuff resulted. As Zlodowski et al. showed in their review, 99% of scapular body fractures are treated conservatively with good to excellent results in 86% of cases [8]. These findings are most likely linked to the fact that the scapular body is enlaced with an extensive muscular zone. Nevertheless, stress on this muscular envelope like rotator cuff dysfunction, scapulothoracic dyskinesis, or impingement can have an impact on the clinical outcome. The presented results of 38% concomitant intra-articular glenohumeral injuries in scapula body Fx demonstrate that injuries to the periscapular muscular zone occur more often than suspected so far. In general, partial or full thickness rotator cuff tendon tear are a common pathology causing pain and impaired shoulder function through all ages. The prevalence of such injuries ranges from 13% to 32% [26] and is strongly correlated to patient age. Sher et al. showed that only 4% of patients younger than 40 years had pathologic findings of the rotator cuff [27]. 

In the last years, several trauma mechanisms resulting in scapula body fractures were described considering blunt high-energy trauma like falls from height and traffic accidents as main causes [3,6]. According to the literature, 50% of scapula body Fx are detected in motorcycle accidents and 20% in pedestrians versus vehicle accidents. Our findings are similar but not totally in line with the literature. Our actual data showed a distribution of trauma with 9 cases of motor vehicles (42.84%) accidents and two (9.5%) pedestrians being traumatized. Regarding falls from height with a reported prevalence of 25–50% [14,28], 9 of the presented patients (42.84)% suffered a fall as well. 

Previous studies lead to the conclusion that scapula body as well as scapula Fx in general are considered so-called “red flag diagnosis” and are associated with concomitant injuries of the head, neck, thorax, and lung as well as with Fx of spine and clavicle [6,9]. In this context, Thompson et al. described an average of four concomitant injuries per patient suffering from scapula Fx [6]. These authors named rib Fx (53.6%), pulmonary contusions (53.6%), and clavicular Fx (26.8%) as the most common concomitant injuries. Lantry et al. showed in their review of 11 studies involving 160 cases that 61.3% of the injured patients had concomitant injuries. This review names rip Fx (32.5%), chest- (19.4%), and head injuries (20%) as the most frequent concomitant injuries [9]. Baldwin et al. retrospectively analyzed a national database and identified 9453 cases of scapular Fx patients. He reported rib fractures in 52.9% of the patients, clavicle Fx in 25.2% of the study population, and in 29.1% of the patients, spinal fractures could be detected. Additionally, pneumothorax was observed in 32.9% of the cases and lung injuries in general were detected in 47.1% [29].

In our study, 95.2% (*n* = 20) of the enrolled patients had at least one concomitant injury of other body parts than the shoulder joint. Compared to Thompson’s analysis with four concomitant injuries per person, in our population, an average of 2.4 additional injuries per patient were identified (0–8) [6]. Multiple rip Fx were as often detected (38.1%, *n* = 8) as traumatic brain injuries (TBI) 38.1% (*n* = 8). Fractures of the skull and facial bones were present in 6 patients (28.6 %). Spinal injuries occurred in 23.8% (*n* = 5). The presented results are similar to the study of Baldwin et al. who showed that rip Fx and thoracic injuries were the major concomitant injuries associated with scapula fractures [29]. 

Due to the fact that 2/3 of all scapula fractures are scapular body Fx [30], only patients with this fracture type were evaluated in this study. In the international literature, the indication for treating scapula body fractures surgically is controversially discussed and poor to excellent results have been shown in both conservative and surgical treatment [31,32]. Different aspects such as outcome, pain, non-union, and surgical approach were analyzed as well as advantages and disadvantages of ORIF in contrast to conservative treatment of scapula body Fx [2,8,28,33,34,35,36]. None of these studies gave thought to the fact that poor outcome, pain, and limited range of motion might result from concomitant intra-articular glenohumeral injuries. To the best of our knowledge, there exists no study in the current literature analyzing the prevalence of concomitant intra-articular glenohumeral injuries in scapula body Fx. However, our study shows that in 8 of 21 enrolled patients, associated injuries of the shoulder joint, in most cases, partial or even complete supraspinatus tendon tears were present. The demonstrated results are in context with the literature. In fact, Beirer et al. focused on concomitant intra-articular glenohumeral injuries in displaced Fx of the lateral clavicle [17]. The authors found in 13 of 28 patients, additional intra-articular injuries during arthroscopy, completing the actual open reduction and internal fixation. In 28.6% of the cases, additional surgical treatment was necessary (*n* = 8). Similar results were shown by Tischer et al. [16]. These authors investigated the incidence of associated injuries in patients with acute acromioclavicular joint dislocations type Rockwood III through V. During the arthroscopic AC-joint repair, concomitant intra-articular injuries were found in 14 of 77 patients (18.2%) and treated during the same operative session. 

## 5. Outcome

In reference to the criteria for indicating ORIF published by Goss et al., in 6 of our patients (28.6%), an operation using a modified Brodsky approach was performed [13,21]. Two of these patients received an additional arthroscopic rotator cuff repair. However, since the number of patients (*n* = 2) treated for additional concomitant injuries is relatively small, no conclusion between the MR-based found and actually treated associated glenohumeral injuries and an outcome respectively can be drawn. Nevertheless, every shoulder surgeon should have additional concomitant intra-articular glenohumeral injuries in mind to rule out as a potential obstacle to a good to excellent clinical outcome when treating scapula body fractures as presented here.

## 6. Limitations 

Despite the fact that the study was designed as a prospective and descriptive work without considering a control group as well as postoperative functional evaluation, several limitations have to be mentioned. First of all, the number of included patients. Incidence of scapula body fractures is rather low, thus since the scapular body Fx is considered rare, the number of 21 enrolled patients can be regarded as adequate and similar compared to previous published studies focusing on scapula body Fx [15,34,36,37].

However, since Fx of the scapular body are a rare injury and the presented study reports, to the best of our knowledge, concomitant intra-articular glenohumeral lesions for the first time, the results are considered as relevant.

Diagnostic arthroscopy is considered the gold standard. Intra-articular glenohumeral lesions can be assessed and addressed during the same operation in one step. However, these considerations do not meet our medical and ethical standards performing arthroscopy of the glenohumeral joint without proven necessity or on the basis of suspicion. In this context, we were able to show that MRI of the glen humeral joint circumstantiated our clinical suspicion.

## 7. Conclusions

In patients suffering from scapula body Fx-associated injuries to the musculoskeletal system, head, and spine are well known and published in the common literature, so when a patient presents with a scapula body fracture, at least three concomitant injuries can be assumed. 

However, traumatic concomitant glenohumeral injuries in scapula body Fx seem to be more frequent than generally expected. Although the overall outcome of scapular body fractures is regarded to range from good to excellent, there are numerous patients with inferior shoulder function. In this context, our study might help to explain these outliers. Subsequent surgical treatment of formerly missed but therapy-relevant injuries may increase functional outcome and reduce the postoperative complication rate following scapula body Fx. Nevertheless, an individual case-based decision is crucial. The indication for ORIF treatment of scapula body fractures must not be misguided by MRI findings. The indication for ORIF has to be made based on the accepted (medial/lateral displacement >20 mm, tilt >45°, displacement >15 mm with tilting >30°, and a glenopolar angle >20°) criteria [22]. In conclusion, we consider our treatment regimen of scapula body Fx including MRI of the shoulder joint as adequate and save way to detect, so far, easily overseen but relevant glenohumeral lesions.

## Figures and Tables

**Figure 1 jcm-09-00943-f001:**
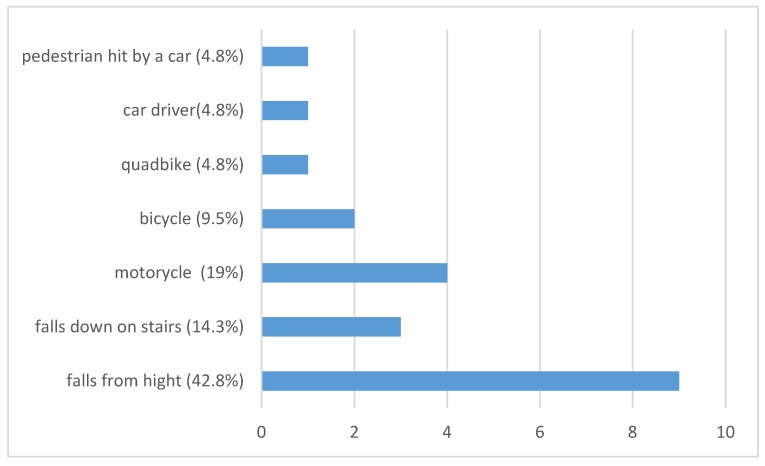
Encountered trauma mechanism which led to scapula fracture.

**Figure 2 jcm-09-00943-f002:**
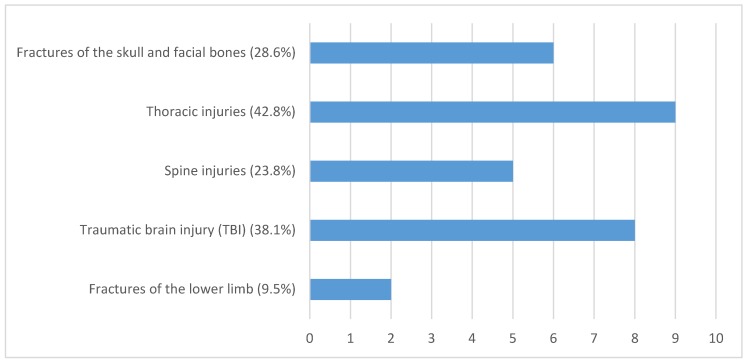
Concomitant injury detected through preoperative X-ray and computed tomography (CT -scan).

**Figure 3 jcm-09-00943-f003:**
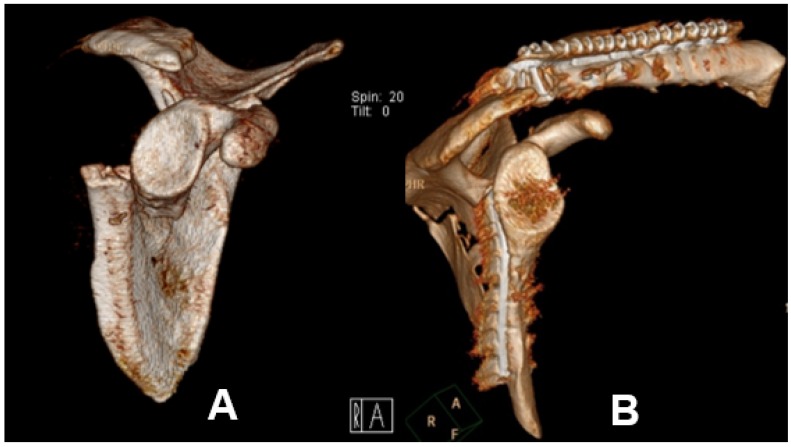
3D reconstruction of CT imaging of displaced scapula body fracture classified according to the orthopedic trauma association as OTA 14.A3.2 (**A**) before Open Reduction Internal Fixation (ORIF), (**B**) after ORIF.

**Figure 4 jcm-09-00943-f004:**
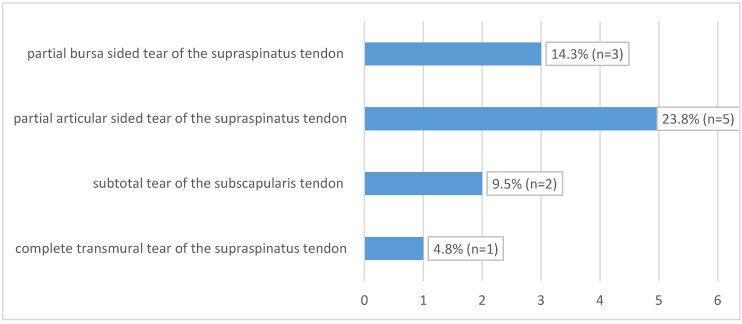
Concomitant intra-articular injuries detected through preoperative magnetic resonance imaging (MRI) and of affected shoulder.

**Figure 5 jcm-09-00943-f005:**
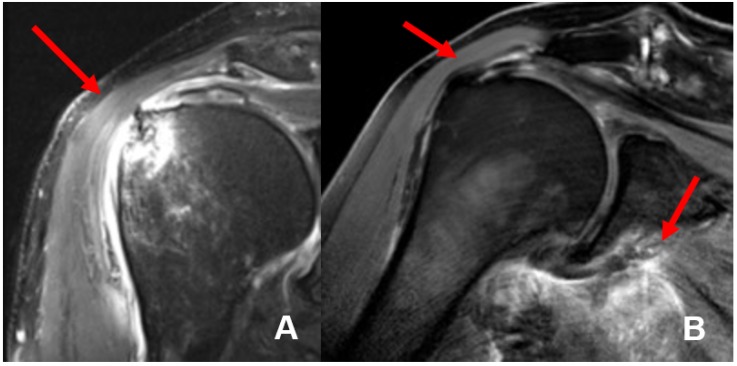
MRI imaging of a complete transmural tear of the supraspinatus tendon (**A**) and complete supraspinatus tendon tear at the foot print with intramuscular hematoma of the musculus subscapularis (**B**) detected in our study.

**Table 1 jcm-09-00943-t001:** Patients’ age and the presence of a concomitant intra-articular injury reveal a weak correlation.

Gender	Age	Concomitant Intra-Articular Glenohumeral Lesion
M	25	no
M	26	no
M	30	no
F	35	no
F	43	no
F	44	no
M	47	partial articular sided tear of the supraspinatus tendon and subtotal tear of the subscapularis tendon
M	48	partial bursa sided tear of the supraspinatus tendon and subtotal tear of the subscapularis tendon
M	50	no
M	54	no
M	56	partial articular sided tear of the supraspinatus tendon
M	56	complete transmural tear of the supraspinatus tendon and partial bursa sided tear of the supraspinatus tendon
M	61	partial articular sided tear of the supraspinatus tendon
M	61	no
M	64	partial bursa sided tear of the supraspinatus tendon
M	64	no
M	64	partial articular sided tear of the supraspinatus tendon
F	66	no
F	70	no
M	71	no
M	83	partial articular sided tear of the supraspinatus tendon
	Pearson’s correlation coefficient	(r) 0.34

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
