# Peer review of "Concomitant Intra-Articular Glenohumeral Lesions in Fractures of the Scapula Body"

_jcm, 2020, doi:10.3390/jcm9040943_

Round 1

Reviewer 1 Report

This is an informative study however; this study was conducted on a small cohort of patients. I would recommend increasing the sample size.

Also, the average age of patient cohort was 53 years, which is another limitation of this study.   Could you please include a table with patient age, gender and the presence/absence of intra-articular glenohumeral injuries.

Author Response

Dear Editor, dear reviewers,

Thank you very much for the kind consideration of our manuscript

Concomitant Intra-articular Glenohumeral lesions in Fractures Of The Scapula Body

and the valuable comments of all reviewers. We revised the manuscript according to your suggestions. Hence it is a privilege to submit a revised version hereby.

Response to Reviewer 1:

  1. This is an informative study however; this study was conducted on a small cohort of patients. I would recommend increasing the sample size.

We totally understand your concern and agree with you. However, we focus on a very rare entity and our prospective study protocol includes the performance of a MRI. Although we conduct our study at an academic level 1 trauma center we are not able to significantly increase our study cohort within a limited time period. In addition the cohort size is comparable with other studies focusing on scapular pathologies.

Maybe a future multicenter study may help to gather additional information regarding our findings.

  1. Also, the average age of patient cohort was 53 years, which is another limitation of this study. Could you please include a table with patient age, gender and the presence/absence of intra-articular glenohumeral injuries.

Thank you for this excellent comment. We added the table with the requested information (see page 6, line 200ff).

Reviewer 2 Report

The paper is well written and easily understanding in each part.

The authors investigated concomitant intra-articular gleno-humeral lesions occurring in scapular body fractures.

Previous studies in literature usually focused on incidence of visceral injuries or other fractures (such as chest fractures or vertebral fracture). To my knowledge, no other studies investigate gleno-humeral lesions in scapular fracture.

Anyway the study has several limitations. The patient group is not so large and heterogeneous in the patients’ age. The data are well collected, but they are defective. There is no correlation between the presence of intra-articular lesions and the age of the patient; it could be a good parameter to address surgeons to perform a more accurate imaging (such as MRI) and, in case, an adequate treatment.

The authors assume that the concomitant rotator cuff tear are acute traumatic because of absence of fatty degeneration or atrophy, but this statement cannot be considered as certainty.

Conservative and surgical treatment of scapular body fracture has good to excellent clinical results (“Outcomes following extra-articular fractures of the scapula: A systematic review” Injury. 2020 Jan 27. Bi AS, Kane LT, Butler BA, Stover MD; “Scapular Body Fracture in the Athlete: A Systematic Review” HSS J. 2018 Oct;14(3):328-332. Neral M, Knapik DM, Wetzel RJ, Salata MJ, Voos JE.), so concomitant surgical treatment of gleno-humeral lesions could result in an overtreatment.

References are exhaustive, but lacking of some recent papers (such as the previous cited ones).

Author Response

Dear Editor, dear reviewers,

Thank you very much for the kind consideration of our manuscript

Concomitant Intra-articular Glenohumeral lesions in Fractures Of The Scapula Body

and the valuable comments of all reviewers. We revised the manuscript according to your suggestions. Hence it is a privilege to submit a revised version hereby.

Response to Reviewer 2:

  1. Anyway the study has several limitations. The patient group is not so large and heterogeneous in the patients’ age. The data are well collected, but they are defective. There is no correlation between the presence of intra-articular lesions and the age of the patient; it could be a good parameter to address surgeons to perform a more accurate imaging (such as MRI) and, in case, an adequate treatment.

Thank you for your valuable comment. Regarding the group size, please refer to answer 1 / Reviewer 1. According to your advice we calculated the correlation of intra-articular lesions and the age of the patient and added dedicated paragraph (see page 6, line 195ff)

  1. The authors assume that the concomitant rotator cuff tear are acute traumatic because of absence of fatty degeneration or atrophy, but this statement cannot be considered as certainty.

            You are absolutely right, the provide information is incomplete and therefore misleading. All MRIs were read by an expert musculoskeletal radiologist as well as an expert shoulder surgeon. Further criteria for evaluation of the acuteness of the injury were indirect signs like bone bruise of the humeral head and joint effusion. Therefore we added the following sentence (see page 5, line 172ff) “Moreover, indirect signs of acute rotator cuff injury was found in terms of 1st bone bruise of the lesser and respectively the greater tuberosity and 2nd joint effusion in terms of acute hematoma.”

  1. Conservative and surgical treatment of scapular body fracture has good to excellent clinical results (“Outcomes following extra-articular fractures of the scapula: A systematic review” Injury. 2020 Jan 27. Bi AS, Kane LT, Butler BA, Stover MD; “Scapular Body Fracture in the Athlete: A Systematic Review” HSS J. 2018 Oct;14(3):328-332. Neral M, Knapik DM, Wetzel RJ, Salata MJ, Voos JE.), so concomitant surgical treatment of glenohumeral lesions could result in an overtreatment.

This is a crucial point. Although the overall outcome is good, the cited review includes a number of studies reporting on Constant scores ranging from 50 to 100, what means inferior to excellent. Facing these data we regard our study to be of significant value for explanation of inferior outcomes following scapular body fractures. However, you are absolutely right detecting every pathology using MRI might induce overtreatment of the patient. Therefore we emphasize the importance of an individual case decision.

We added a paragraph regarding this topic to the conclusion section (page 9, line 309ff).

  1. References are exhaustive, but lacking of some recent papers (such as the previous cited ones).

            We added the required citations.

We hope that our revised manuscript is now considerable for publication in Journal of Clinical Medicine.

Sincerely

Michael Zyskowski

Round 2

Reviewer 2 Report

Thank you for considering my notes. I appreciated your changes in the paper.
I think it's crucial to enphasize that your finding don't change indications in scapular body fracture treatment, in order to don't create mistakes. The paragraph added to the conclusion section (page 9, line 309ff) clarifies that an individual case-based decision is crucial, but I would stress that MRI study has not to be a parameter to guide our decision in scapular body fractures treatment, in acute setting.

Author Response

Dear Editor, dear reviewers,

Thank you very much for the kind consideration of our manuscript

Concomitant Intra-articular Glenohumeral lesions in Fractures Of The Scapula Body

and the valuable comments of all reviewers. We revised the manuscript according to your suggestions. Hence it is a privilege to submit a revised version hereby.

Response to Reviewer 2:

  1. Thank you for considering my notes. I appreciated your changes in the paper.
    I think it's crucial to emphasize that your finding don't change indications in scapular body fracture treatment, in order to don't create mistakes. The paragraph added to the conclusion section (page 9, line 309ff) clarifies that an individual case-based decision is crucial, but I would stress that MRI study has not to be a parameter to guide our decision in scapular body fractures treatment, in acute setting.

We totally understand your concern and agree with you it is crucial to point out that the indication for ORIF of the scapula must not be mislead by MRI findings and has to be stated based on accepted criteria, like those published by Cole et al [1].

We added a dedicated sentence to the conclusion section (page 9, line 314ff).

We hope that our revised manuscript is now considerable for publication in Journal of Clinical Medicine.

Sincerely

Michael Zyskowski

  1. Cole, P.A., et al., Radiographic follow-up of 84 operatively treated scapula neck and body fractures. Injury, 2012. 43(3): p. 327-33.